# Access and enrollment in safety net programs in the wake of COVID-19: A national cross-sectional survey

**Brendan Saloner[1]\*, Sarah E. Gollust[2], Colin Planalp[3], Lynn A. Blewett[2,3]**

**1** Department of Health Policy and Management, Johns Hopkins Bloomberg School of Public Health, Baltimore, Maryland, United States of America, **2** Division of Health Policy and Management, University of Minnesota School of Public Health, Minneapolis, Minnesota, United States of America, **3** State Health Access Data Assistance Center (SHADAC), University of Minnesota School of Public Health, Minneapolis, Minnesota, United States of America

\* bsaloner@jhu.edu

**Data Availability Statement:** All relevant data are available at: https://osf.io/54673/.

**Funding:** The authors received no specific funding for this work.

## Abstract

The global COVID-19 pandemic is causing unprecedented job loss and financial strain. It is unclear how those most directly experiencing economic impacts may seek assistance from disparate safety net programs. To identify self-reported economic hardship and enrollment in major safety net programs before and early in the COVID-19 pandemic, we compared individuals with COVID-19 related employment or earnings reduction with other individuals. We created a set of questions related to COVID-19 economic impact that was added to a cross-sectional, nationally representative online survey of American adults (age ≥18, English-speaking) in the AmeriSpeak panel fielded from April 23–27, 2020. All analyses were weighted to account for survey non-response and known oversampling probabilities. We calculated unadjusted bivariate differences, comparing people with and without COVID-19 employment and earnings reductions with other individuals. Our study looked primarily at awareness and enrollment in seven major safety net programs before and since the pandemic (Medicaid, health insurance marketplaces/exchanges, unemployment insurance, food pantries/free meals, housing/renters assistance, SNAP, and TANF). Overall, 28.1% of all individuals experienced an employment reduction (job loss or reduced earnings). Prior to the pandemic, 39.0% of the sample was enrolled in ≥1 safety net program, and 50.0% of individuals who subsequently experienced COVID-19 employment reduction were enrolled in at least one safety net program. Those who experienced COVID-19 employment reduction versus those who did not were significantly more likely to have applied or enrolled in ≥1 program (45.9% versus 11.7%, p<0.001) and also significantly more likely to specifically have enrolled in unemployment insurance (29.4% versus 5.4%, p < .001) and SNAP (16.8% versus 2.8%, p = 0.028). The economic devastation from COVID-19 increases the importance of a robust safety net.

**Competing interests:** The authors have declared that no competing interests exist.

## Introduction

The United States is at the center of the global COVID-19 pandemic, with more than one-quarter of the world's confirmed cases and one-fifth of all deaths as of mid-August 2020 [1]. Along with surging mortality rates, COVID-19 has brought economic devastation to many American families. Beginning in March 2020, most states enacted mandatory shelter-in-place orders to reduce transmission and many businesses closed, leading to decreased economic activity and widespread layoffs. From March to April 2020, an estimated 20.5 million individuals became unemployed and the unemployment rate reached 14.7%, rebounding to 10.2% in July 2020 [2].

National surveys from early in the pandemic highlight the financial strain faced by US families. A survey conducted from April 15–20, 2020, found that 31.0% of households experienced difficulties affording basic needs or paying bills [3]. One-fifth of households experienced inadequate access to food in April 2020, with higher rates among families with young children [4]. The burden of these challenges is falling disproportionately on people who are part-time workers, have children, are younger, or are racial/ethnic minorities [5, 6]. While economic activity resumed in many states in May 2020, the resurgence of the epidemic in many parts of the U.S. in the summer prompted a new wave of business and school closures. The Census Bureau's Pulse Household Survey documented that family hardship–including food scarcity and housing insecurity–trended upward from April 23 to July 21, 2020 (the most recently available data for the current study) [7].

Many individuals turned to safety net programs for assistance early in the pandemic. News media reports have profiled the growing demand on food pantries and social assistance programs [8, 9]. Unemployment insurance programs, which are a joint federal-state partnership to support displaced workers, are also seeing enormous demand [10]. Means-tested, federally-funded programs such as Medicaid, the Supplemental Nutrition Assistance Program (SNAP), and Temporary Assistance for Needy Families (TANF) are expected to provide some assistance for low-income individuals. However, resources for programs like TANF are limited due to their block-grant structure, income eligibility, and work requirements [11]. Assistance programs were given a small one-time boost in the $2.0 trillion Coronavirus Aid, Relief, and Economic Security (CARES) Act, which extended the duration of unemployment insurance benefits and provided $1,200 in direct payment to eligible adults and $500 to dependent children [12]. In late August 2020, with federal unemployment assistance set to expire, the US Congress was at an impasse over the scope of a potential relief measure. While Democrats were pushing for a renewal of the $600 weekly unemployment insurance payments, the Trump Administration and some Congressional Republicans resisted a more extensive relief package [13].

As Congress and the states contemplate further investments in safety net programs, there is an urgent need to identify how families affected by COVID-19 have accessed available programs before and since the pandemic and the challenges they expected to confront. Further, given the substantial economic burden of the COVID-19 pandemic on already disadvantaged groups, it is important to identify those most in need in order to better target safety net funding and program activities, including outreach and enrollment. Further, data from early in the pandemic provides an important baseline for evaluating the evolving changes in program participation and hardship among vulnerable individuals. We therefore conducted a nationally representative survey regarding the COVID-19-related hardships experienced by American adults and their use of safety net services early in the pandemic, and hypothesized that the impact of COVID-19 and enrollment in safety net programs would be greatest among those experiencing COVID-19-related employment and earnings loss.

## Methods

We collected data using the AmeriSpeak Omnibus survey, a nationally representative panel of American households recruited and maintained by NORC at the University of Chicago. The Omnibus survey is a cross-sectional survey of a rotating set of the AmeriSpeak panel that is conducted every two weeks and to which researchers and other partners can contribute items. The AmeriSpeak panel is recruited using stratified, address-based sampling methods that cover approximately 97.0% of all residential addresses. The multi-stage probability sample is created using a national frame area where blocks are sampled from within defined metropolitan or rural areas. AmeriSpeak oversamples in areas with a higher concentration of young adults and minorities and engages in additional efforts to follow up with households that initially do not respond. Individuals are recruited to the panel using a combination of US mail, telephone interviews, and in-person field interviews. Households can respond to the survey by internet (including on smartphones) or by telephone interview. About 85% of the interviews are completed online and 15% are conducted over the phone. The phone option is offered to allow "net-averse" households to participate. The overall response rate for the panel is about 34.0% (American Association for Public Opinion Research [AAPOR] response rate three) [14].

For this study, our team developed the State Health Access Data Assistance Center (SHA-DAC) COVID-19 Safety Net Survey and contracted with NORC to add the survey questions to the survey that was in the field April 23 to April 27, 2020. We contracted with NORC to administer the survey to a target of 1,000 respondents. The study was restricted to people over age 18. The final sample included 1,007 adults. Table 1 shows the demographic characteristics of the study sample. NORC develops weights to national census benchmarks and balances by gender, age, education, race/ethnicity, and region. The weighted sample is similar to a national sample of adults: 51.4% of the sample was female, 44.8% between age 18 and 44, 37.4% non-white, 46.1% with a chronic condition, 36.2% with a high school degree or less, and 83.8% residing in metropolitan areas.

The current study analyzes unemployment reduction, economic burden, and use of government safety net and assistance programs. The main comparison of interest is between

**Table 1. Demographics of study sample.**

|  | Unweighted n | Weighted % | Standard Error |
|---|---|---|---|
| Female | 524 | 51.4 | 2.04 |
| Age Group |  |  |  |
| 18–29 | 113 | 18.1 | 1.94 |
| 30–44 | 268 | 26.7 | 1.78 |
| 45–59 | 251 | 24.5 | 1.66 |
| 60+ | 375 | 30.7 | 1.75 |
| Non-white | 362 | 37.4 | 2.00 |
| Any chronic condition | 492 | 46.1 | 2.05 |
| Education |  |  |  |
| No HS diploma | 36 | 8.8 | 1.53 |
| HS graduate or equivalent | 126 | 27.5 | 2.13 |
| Some college | 293 | 28.5 | 1.72 |
| BA or above | 552 | 35.3 | 1.70 |
| Resides in a metro area | 885 | 83.8 | 1.65 |

**Note:** Weighting performed using survey weight created by NORC to approximate to national proportions. Sample size = 1,007 individuals.
**Source:** Authors' analysis of the April 2020 SHADAC COVID-19 survey.

individuals who experienced either job loss or loss of income from COVID-19 (which we call "employment reduction") versus those who did not (see Table 2 for categories). We examined whether individuals in both groups were aware of seven major safety net programs (Medicaid, health insurance marketplaces, unemployment insurance, food pantries/free meals, housing/renters assistance, SNAP, and TANF). We further examined whether individuals said that they were enrolled before the pandemic in each of these programs and whether they had enrolled or applied since the pandemic. We also asked individuals how they intended to spend their $1,200 stimulus checks to assess spending priorities. Finally, we asked individuals to rate their confidence in their ability to pay for several basic needs and expenses over the next four weeks from date of the survey. The items that were used in the survey were developed by our team for the purpose of this study; items were not piloted before being used in the study.

Each of our main outcomes was dichotomized. Survey weights were used in all analyses to account for known differences in sampling probability and non-response. In bivariate analysis, we compared individuals with and without COVID-19-related employment or earnings loss on their safety net awareness and participation, plans for spending stimulus checks, and confidence in ability to pay for basic needs. We calculated two-sided $t$-tests for differences in means between the groups. Because we were interested in identifying the disproportionate changes in program participation among those experiencing employment reduction, we fit a regression model for program participation that estimates the average change since the pandemic, the baseline rate for those without employment reduction, and an interaction term. This interaction term is analogous to a difference-in-differences coefficient, representing the change in program participation since the pandemic for those with employment reduction versus those who without employment reduction. The study was determined exempt by the University of Minnesota Institutional Review Board (IRB). [Study data will be archived at YYY DOI number ZZZ after paper acceptance].

## Results

Overall, 28.1% of respondents said that they experienced an employment or earning loss due to the coronavirus, with 10.6% specifically reporting losing a job, 12.3% reporting losing work hours, and 5.3% reporting a pay cut (Table 2).

**Table 2. Experience of employment or earnings loss related to COVID-19.**

| Category | Percent | Standard Error |
|---|---|---|
| Coronavirus Employment or Earnings Loss | 28.1% | 1.9% |
| I have had my work hours cut due to the coronavirus | 12.3% | 1.3% |
| I have lost my job due to the coronavirus | 10.6% | 1.5% |
| I have had my pay cut due to the coronavirus | 5.3% | 0.8% |
| I have retired from work due to the coronavirus | 1.3% | 0.5% |
| I am on paid leave because my employer closed due to the coronavirus | 2.4% | 0.5% |
| No Coronavirus Employment or Earnings Loss | 71.0% | 1.9% |
| I was not employed at the onset of the coronavirus | 28.3% | 1.8% |
| I am working from home due to the coronavirus | 17.3% | 1.4% |
| I have gotten a new job because of the coronavirus | 0.3% | 0.3% |
| I am working more hours due to the coronavirus | 7.7% | 1.2% |
| The coronavirus has not affected my job | 20.4% | 1.7% |

**Note:** Individuals can identify more than one factor for employment status so categories sum to greater than 100%.
**Source:** Authors' analysis of the April 2020 SHADAC COVID-19 survey.

**Table 3. Awareness and enrollment in safety net programs before and since pandemic.**

| Program | Awareness of program | Enrollment in the Program | | | |
|---|---|---|---|---|---|
| | | Prior to the Pandemic | Since the Pandemic | Change in Enrollment | P-Value for Change |
| At Least One Safety Net Program | 98.00% | 39.00% | 46.72% | 15.29 | p<0.001 |
| Medicaid | 91.90% | 21.90% | 23.61% | 1.70 | p<0.001 |
| Health insurance exchanges | 71.20% | 11.30% | 13.11% | 1.80 | 0.001 |
| Unemployment insurance | 77.70% | 9.20% | 17.05% | 7.87 | p<0.001 |
| Food pantry/free meals | 89.20% | 11.50% | 14.86% | 3.33 | p<0.001 |
| Housing/renters assistance | 69.80% | 5.70% | 6.72% | 1.07 | 0.009 |
| SNAP | 95.60% | 20.10% | 24.32% | 4.27 | p<0.001 |
| TANF | 63.80% | 1.70% | 2.55% | 0.90 | 0.009 |

**Note:** P-value represents a t-test for the change in enrollment prior to the pandemic and since the pandemic. Change in enrollment is in percentage points.

**Source:** Authors' analysis of the April 2020 SHADAC COVID-19 survey.

Respondents' awareness and enrollment in seven major safety net programs before and after the pandemic is recorded in Table 3. Overall, a majority reported that they were aware of each program, with the highest awareness for SNAP (95.6%), Medicaid (91.9%), and food pantries (89.2%) and the lowest awareness reported for housing/renters assistance (69.8%) and TANF (63.8%). Correspondingly, overall program participation was highest prior to COVID-19 in Medicaid (21.9%) and SNAP (20.1%). Prior to the pandemic, 39.0% of the sample was enrolled in at least one safety net program. Overall, there were significant increases in the percent of people reporting that they had applied for or enrolled in most safety net programs since the pandemic. The overall percentage reporting at least one program increased by 15.29 percentage points (p<0.0001), with the largest reported changes for unemployment insurance (7.87 percentage points, p<0.0001) and SNAP (4.27 percentage points, p<0.0001).

Table 4 disaggregates the safety net program participation data before and since the pandemic between people who did versus did not experience employment reduction. Participation in at least one safety net program prior to the pandemic was higher among people who

**Table 4. Comparing changes in program participation among individuals who experienced employment reduction versus those who did not.**

| | Prior to the Pandemic | | Since the Pandemic | | Change in Enrollment (Difference)* | | Diff-in-Diff* | p-value for Diff-in-diff |
|---|---|---|---|---|---|---|---|---|
| | ER | NER | ER | NER | ER | NER | | |
| At Least One Safety Net Program | 50.00% | 37.70% | 77.50% | 43.08% | 27.57 | 5.37 | 22.20 | 0.038 |
| Medicaid | 29.70% | 21.00% | 36.31% | 22.11% | 6.60 | 1.10 | 5.50 | 0.596 |
| Health insurance exchanges | 23.00% | 10.00% | 30.62% | 11.10% | 7.60 | 1.10 | 6.50 | 0.505 |
| Unemployment insurance | 5.30% | 9.60% | 34.65% | 14.99% | 29.40 | 5.40 | 24.00 | p<0.001 |
| Food pantry/free meals | 12.20% | 11.50% | 16.54% | 14.67% | 4.40 | 3.20 | 1.20 | 0.882 |
| Housing/renters assistance | 5.20% | 5.70% | 10.37% | 6.30% | 5.10 | 0.60 | 4.50 | 0.335 |
| SNAP | 35.30% | 18.30% | 52.07% | 21.06% | 16.80 | 2.80 | 14.00 | 0.201 |
| TANF | 3.10% | 1.50% | 5.83% | 2.16% | 2.70 | 0.70 | 2.00 | 0.671 |

**Notes:** "ER" = COVID-19 related employment reduction, "NER" = no COVID-19 related employment reduction. Diff-in-diff represents the change in enrollment since the pandemic for ER group relative to the NER and is estimated from a regression model that includes an interaction between "post" pandemic and being in the ER group. P-value for diff-in-diff is the p-value associated with that interaction term.

*Unit reported is a percentage point change.

**Source:** Authors' analysis of the April 2020 SHADAC COVID-19 survey.

**Table 5. Priority for stimulus check spending.**

|  | Everyone | COVID Job Loss | No COVID Job Loss | P-Value |
|---|---|---|---|---|
| Mortgage or rent | 24.2% | 47.1% | 21.6% | 0.003 |
| Utilities (electricity, water, heat, gas, internet, etc.) | 17.6% | 17.2% | 17.6% | 0.937 |
| Food for myself/family | 13.6% | 12.9% | 13.6% | 0.888 |
| Credit card debt, car payments, student loans | 16.3% | 5.8% | 17.5% | 0.003 |
| Medical care or insurance premiums | 3.7% | 0.0% | 4.1% | p<0.001 |
| Savings or Investment | 10.3% | 9.1% | 10.4% | 0.873 |
| Donation | 2.8% | 0.0% | 3.1% | p<0.001 |

**Note**: We combined response categories for paying off credit card debt, making a car payment, and paying off student loans and we combined response categories for paying for medical care already received, needed medical care, and insurance premiums.

**Source:** Authors' analysis of the April 2020 SHADAC COVID-19 survey.

subsequently experienced COVID-19 employment reduction than those not experiencing employment reduction (50.0% versus 37.7%). Individuals experiencing employment reduction were more likely to have participated in the health insurance exchanges (23.0% versus 10.0%) and SNAP (35.3% versus 18.3%) before the pandemic than those who did not. The difference-in-differences coefficient, which identifies the relative difference in program participation since the pandemic between the two groups shows that those in the employment reduction group were significantly more likely to have enrolled in at least one program since the pandemic (22.2 percentage point increase, p = 0.038). The increase was particularly notable for increased participation in unemployment insurance (24.0 percentage point increase, p = 0.0009).

The highest priority for stimulus check spending among the provided categories was mortgage/rent (24.2%), followed by utilities (17.6%), debt and loans (16.3%), food (13.6%), savings or investment (10.3%), medical care and insurance (3.7%), and donations (2.8%) (Table 5). Compared to those with no employment reduction, people experiencing COVID-19 employment reduction were significantly more likely to plan using the checks for mortgage/rent (47.1% versus 21.6%, p = 0.003) and less likely to use it for debt (5.8% versus 17.5%, p = 0.003), donations (0.0% versus 2.0%, p<0.001), and medical expenses (0.0% versus 4.1%, p<0.001).

People experiencing COVID-19-related employment reduction were significantly less likely to report being confident in their ability to pay all categories of basic expenses (Table 6). Overall, 33.5% of the sample was not confident in their ability to pay for at least one type of expense, with 69.9% of those with COVID-19 employment reduction reporting a lack of confidence in

**Table 6. Not confident in ability to pay for basic needs over next four weeks.**

| Program | Everyone | COVID Job Loss | No COVID Job Loss | P-Value |
|---|---|---|---|---|
| Not confident in at least one of the options below | 33.5% | 69.9% | 28.8% | p<0.001 |
| Mortgage or rent | 12.8% | 35.8% | 9.7% | 0.002 |
| Utilities (electricity, water, heat, gas, internet, etc.) | 9.3% | 34.8% | 6.4% | p<0.001 |
| Food for myself/family | 6.9% | 32.6% | 3.9% | p<0.001 |
| Credit card debt, car payments, student loans | 21.2% | 55.8% | 17.2% | p<0.001 |
| Medical care or insurance premiums | 18.7% | 45.3% | 15.5% | p<0.001 |

**Note**: Table reports on percentages responding that they were either "not at all" or "not very" confident in their ability to pay this expense. We combined response categories for paying off credit card debt, making car payment, and paying off student loans and we combined response categories for paying for medical care already received, needed medical care, and insurance premiums.

**Source:** Authors' analysis of the April 2020 SHADAC COVID-19 survey.

ability to pay expenses versus 28.8% of those with no COVID-19 employment reduction (p<0.001). Our analysis found significant differences between the two groups (employment reduction versus no employment reduction) in every category of expense. Confidence was lowest for ability to pay debt (55.8% versus 17.2%, p<0.001), medical care or insurance premiums (45.3% versus 15.5%, p<0.001), and mortgage or rent (35.8% versus 9.7%, p = 0.002).

## Discussion

This study examined the economic challenges confronting American adults in mid-April 2020, with a focus on individuals experiencing employment or earning loss related to the COVID-19 pandemic. Overall, our study findings highlight the precarity of individuals who were most affected early in the COVID-19 pandemic. While majorities of the sample were aware of safety net programs, awareness was lowest for TANF and the health insurance exchanges. We found that 45.9% of individuals experiencing COVID-19 job loss had enrolled or applied for a safety net program since the pandemic, with the highest application rates reported for unemployment insurance and SNAP. A majority of those experiencing COVID-19-related employment or earnings loss expressed concern about paying for basic needs in the month, especially the ability to pay off existing debt. Importantly, the group that experienced COVID-19-related employment reductions also tended to be the group more likely to have accessed safety net programs before the pandemic, suggesting their heightened vulnerability.

Our findings underscore concerns that have been raised about the sustainability of existing safety net programs in attempting to deal with this extraordinary public health and economic challenge to the U.S. population [15]. They also reveal gaps in knowledge of key safety net programs that could help provide continuity of insurance or benefits to those most in need of such programs. For example, roughly one-quarter of respondents did not have awareness of unemployment insurance; however, these knowledge gaps could be remedied through efforts to increase education and outreach to those who experience job loss. Furthermore, individuals who may have a general awareness of programs may not necessarily know whether they might be eligible. For example, not all workers are eligible for unemployment insurance. To qualify, individuals must acquire a sufficient work history, and the program often excludes certain types of workers (e.g., "gig" workers such as Uber drivers) [10]. Most other safety net programs include large exclusions that limit their reach to vulnerable individuals.

The stability of food assistance programs is of particular concern. The rising levels of food insecurity in the pandemic have been accompanied by a food production crisis. Early in the pandemic, some farmers were resorting to destroying agricultural products they were unable to sell [16]. Expanding food assistance programs could help address food insecurity and the food production crisis. Awareness of SNAP and food pantries were relatively high in our sample. Recent enrollment in SNAP was particularly high, especially among people with COVID-19 employment or earnings loss. However, challenges loom for SNAP enrollees. COVID-19 has delayed, but not eliminated, a recent push by the Trump Administration to increase the stringency of work requirements for SNAP enrollees. While planned work requirements were suspended by the CARES Act, these work requirements would be reinstated as soon as the official national emergency expires, but likely long before the economy sufficiently rebounds. Nearly 700,000 Americans are set to lose benefits under the current requirements, and this number is likely to grow with rising unemployment [17].

The affordability of housing is another concern. Almost half of all individuals experiencing COVID-19 employment or earnings loss said that they would spend their $1,200 stimulus checks on rent or mortgage. However, this monetary infusion offered only small and temporary relief, as median rent in the U.S. is around $1,000 per month [18]. Most states opted to

suspend evictions in the context of state-declared COVID-19 emergencies, but the scope of these protections varied widely across the states [19]. States began lifting these restrictions in May 2020, leading to a resumption in eviction [20]. The CARES Act did include six months of relief from eviction for individuals paying federal mortgages and provided limited funds for community block grants and the federal Housing and Urban Development (HUD) Administration [21]. However, these provisions were unlikely to reach many of the most precarious families, including renters not living in subsidized housing.

As millions have lost their employer-sponsored health insurance it is important to consider the potential impact on public insurance programs. Programs created or expanded under the Affordable Care Act (ACA)—Medicaid and the health insurance exchanges—likely absorbed some of the new demand. In the 37 states (including the District of Columbia) that expanded Medicaid, the program now covers most individuals up to 138% of the Federal Poverty Level (FPL), while the exchanges provide slide-scale subsidies to individuals with household incomes between 100% and 400% FPL. Further, limited emergency Medicaid coverage can be extended to uninsured individuals for COVID-19-related care (including in states that have not expanded Medicaid) [22]. In our sample, 6.6% of people experiencing COVID-19 unemployment reduction indicated they had applied or enrolled in Medicaid since the pandemic and 7.6% had said the same about the insurance exchanges.

The increased enrollment in public programs could have important implications for state budgets and delivery systems. Early in the pandemic, state Medicaid programs began projecting that the weakened economy would increase program enrollment and total spending [23]. Testing and treatment related to COVID-19 could also contribute to rising Medicaid spending, though this spending could be offset by other forms of medical care that decreased after the pandemic, such as preventive office visits. State programs have a variety of policy options for easing transitions of new members into Medicaid, reducing churn, and simplifying enrollment. A recent analysis suggests that Medicaid enrollment increased early in the pandemic, although these increases were not correlated with enrollment changes in those states [24]. Similarly, some states have used their exchanges to provide special COVID-19 open enrollment period, with eligibility open to the currently uninsured due to the pandemic [25]. For those relying on the federal marketplace (healthcare.gov), no such special COVID-19 open enrollment period has thus far been implemented. Both the federal and state exchanges provide for the possibilities of a special enrollment period, but only for a qualified coverage loss, such as the termination of employer-sponsored coverage. States are also taking varying approaches in offering grace periods for non-payment of premiums and in offering special coverage of COVID-19 related services [26].

It is important to consider how individuals who are enrolling in safety net programs are accessing services during a pandemic. Given the closure of many places of business, combined with individuals' potential hesitancy to seek in-person services based on their own perceived coronavirus risk or caregiving needs, individuals may be challenged to visit social services agencies for enrollment or customer service. While some customer services were expanded by phone and online, there have been widespread challenges enrolling in programs like unemployment insurance [27]. Lower Medicaid enrollment in some states may also reflect the challenges of navigating remote eligibility and enrollment processes since the pandemic.

Our study is subject to several limitations. First, the COVID-19 pandemic is a rapidly evolving situation and data collected in late-April 2020 may not generalize to issues and concerns arising in more recent periods. Since April, the pandemic spread widely across the United States, including to many southern and Midwestern states with relatively weaker safety nets. However, data from April provides an important baseline to assess the evolving need of affected Americans, and to ultimately assess economic recovery. Second, the size of our survey

sample (N = 1,007) limits our ability to examine specific subgroups, such as individuals residing in communities that are COVID-19 hotspots. It also limits our statistical power to detect potentially important differences across subgroups. Future research, with larger sample sizes, could beneficially compare differences in program participation based on residence in areas experiencing higher COVID-19 case rates [28]. Third, as with all surveys on public program participation, there is likely to be some under-identification of program participation [29]. Self-reported program participation may be particularly problematic for some programs such as SNAP [30], although we have no reason to believe that reporting bias will be differential between people who experienced COVID-19 employment loss versus those that did not. Fourth, we only asked about past and current use of safety net programs. While asking about future enrollment may be more informative for projecting the burden on the safety net, future intentions are not perfectly predictive of actual enrollment, because of considerable constraints all people face in changing behavior, particularly when considering engaging with complex public programs [31]. Fifth, the data are collected from a nationally representative household panel of English-speaking survey respondents with a modest sample size and thus the study does not offer insights into populations who may be most adversely affected by COVID-19, such as people with limited English proficiency and those experiencing homelessness, in long-term care institutions, and in jails or prisons.

## Conclusion

COVID-19 has created historic public health and economic crises. Our survey highlights the hardship experienced by Americans who experienced loss of employment and income early in the pandemic. The crisis has also revealed the cracks in safety net programs, which have been largely underfunded. While some emergency funding has been provided in the wake of COVID-19, the cumbersome and state-specific mechanisms to assess eligibility and facilitate enrollment have created bottlenecks and long wait times across the country. We also identified gaps in the knowledge of existing safety net programs and a need for targeted funding for outreach and enrollment activities. These programs are critical in affording assistance to those most in need to provide for basic needs of food, housing, and income support. A strong economic recovery will crucially depend on how effectively these programs can be streamlined and sustained during this difficult time.

## Supporting information

**S1 File.**
(DOCX)

## Author Contributions

**Conceptualization:** Brendan Saloner, Sarah E. Gollust, Colin Planalp, Lynn A. Blewett.

**Project administration:** Colin Planalp.

**Writing – original draft:** Brendan Saloner.

**Writing – review & editing:** Sarah E. Gollust, Colin Planalp, Lynn A. Blewett.

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
