## [Decision Letter · Decision Letter 0]

29 Jul 2020

PONE-D-20-20058

Access and enrollment in safety net programs in the wake of COVID-19: A national cross-sectional survey

PLOS ONE

Dear Dr. Saloner,

Thank you for submitting your manuscript to PLOS ONE. After careful consideration, we feel that it has merit but does not fully meet PLOS ONE’s publication criteria as it currently stands. Therefore, we invite you to submit a revised version of the manuscript that addresses the points raised during the review process.

In particular, please address reviewer concerns regarding presentation of data in the Tables, especially Table 2 which is very difficult to read/interpret.  Regarding concerns about generalizability expressed by Reviewer 2, while the authors certainly cannot control the length of time submissions are under review, perhaps the question of generalizability (or lack thereof) could be framed within a broader context.  Finally, further discussion about challenges accessing safety net services during a pandemic could be more fully fleshed out in the revised manuscript. 

We look forward to receiving your revised manuscript.

Kind regards,

Nickolas D. Zaller

Academic Editor

PLOS ONE

Journal Requirements:

2. Please include additional information regarding the survey or questionnaire used in the study and ensure that you have provided sufficient details that others could replicate the analyses. For instance, if you developed a questionnaire as part of this study and it is not under a copyright more restrictive than CC-BY, please include a copy, in both the original language and English, as Supporting Information. Moreover, please include more details on how the questionnaire was pre-tested, and whether it was validated; and clearly report the number of respondents and the response rate.

3. Please correct your reference to "p=0.000" to "p<0.001" or as similarly appropriate, as p values cannot equal zero.

4. In your Methods section, please provide additional information about the participant recruitment method and the demographic details of your participants. Please ensure you have provided sufficient details to replicate the analyses such as: a) the recruitment date range (month and year), b) a description of any inclusion/exclusion criteria that were applied to participant recruitment, c) a table of relevant demographic details, d) a statement as to whether your sample can be considered representative of a larger population, e) a description of how participants were recruited, and f) descriptions of where participants were recruited and where the research took place.

5.We note that you have stated that you will provide repository information for your data at acceptance. Should your manuscript be accepted for publication, we will hold it until you provide the relevant accession numbers or DOIs necessary to access your data. If you wish to make changes to your Data Availability statement, please describe these changes in your cover letter and we will update your Data Availability statement to reflect the information you provide.

6.PLOS requires an ORCID iD for the corresponding author in Editorial Manager on papers submitted after December 6th, 2016. Please ensure that you have an ORCID iD and that it is validated in Editorial Manager. To do this, go to ‘Update my Information’ (in the upper left-hand corner of the main menu), and click on the Fetch/Validate link next to the ORCID field. This will take you to the ORCID site and allow you to create a new iD or authenticate a pre-existing iD in Editorial Manager. Please see the following video for instructions on linking an ORCID iD to your Editorial Manager account: https://www.youtube.com/watch?v=_xcclfuvtxQ

<h1>** **</h1>

Reviewers' comments:

Reviewer's Responses to Questions

**Comments to the Author**

1. Is the manuscript technically sound, and do the data support the conclusions?

Reviewer #1: Yes

Reviewer #2: Yes

2. Has the statistical analysis been performed appropriately and rigorously? 

Reviewer #1: Yes

Reviewer #2: Yes

3. Have the authors made all data underlying the findings in their manuscript fully available?

Reviewer #1: Yes

Reviewer #2: Yes

4. Is the manuscript presented in an intelligible fashion and written in standard English?

Reviewer #1: Yes

Reviewer #2: Yes

5. Review Comments to the Author

Reviewer #1: This study reports the results of a national survey conducted in April 2020 to assess the economic conditions of individuals with job loss related to the COVID-19 pandemic. This is a remarkably fast turnaround for such data and the authors are to be commended for their ability to implement these questions in the field on such a short timeline. The results are interesting, but there presentation could be clearer, and there are interesting discussion points that were not address.

Major comments:

1. Table 2 is the potentially most impactful table in the manuscript but it is difficult to interpret. I would recommend three sections in the table: 1) Enrolled before the pandemic; 2) Enrolled after the pandemic; 3) Change in enrollment. As it currently stands, it’s very difficult to understand whether the enrolled before and applied/enrolled since the pandemic are additive.

2. Discussions points: Even before the pandemic, those who experienced job loss were more likely to need help from a safety net program. This speaks to the vulnerability of those impacted by COVID-19 and should be highlighted. Further, only 6.6% of individuals with job loss enrolled in Medicaid. I would expect it to be higher during a public health crisis and may speak to serious challenges in accessing public services during the pandemic.

3. The last sentence of abstract could be interpreted as individuals are “stretching the safety net programs” but this is what the programs were designed to do. Rephrase to indicate that safety net systems will remain critical until the pandemic is over and should be protected from funding cuts. Please correct similar language in the discussion. For example: “Food assistance programs are of particular concern” - these programs not “of concern” but I do agree that their stability is imperative. The problem is not that they are being used - it’s that the US didn’t develop the programs in a way that could respond to large recessions/depressions. A well run program would be able to support individuals during downturns in the economic cycle.

Minor comments:

1. Rather than comparing those with job loss to those without job loss who enrolled, would be more interesting to highlight how much those with job loss contributed to increases in use of safety net services. This could be done with a simple diff-in-diff model to determine how much of the increase in safety services pre and during COVID was associated with job loss.

2. To build on this comment, comparisons between those with and without covid job loss are not wrong but are somewhat odd. Of course those who lost their job would rely on safety net health systems more than individuals who did not lose their jobs. If the authors believe this is the most appropriate comparison in the data, then a clear justification for why such a comparison is important is warranted.

3. Lines 147-152 - The first part of the sentence refers to those with and without job loss. Unclear whether the later part of the sentence related to general population or only those with job loss. Please clarify.

4. There is substantial commentary on Medicaid in the discussion, but would shorten and refocus on what can be gleaned from the study’s findings. For example, what does it mean for budgets, health care delivery, health system budgets, etc that millions more individuals will be relying on Medicaid?

Reviewer #2: I uploaded my review as an attachment rather than pasting it here.

6. PLOS authors have the option to publish the peer review history of their article (what does this mean?). If published, this will include your full peer review and any attached files.

Reviewer #1: No

Reviewer #2: **Yes: **Charles J. Courtemanche

---

## [Author Response · Author response to Decision Letter 0]

16 Aug 2020

Revision Memorandum: Access and enrollment in safety net programs in the wake of COVID-19: A national cross-sectional survey (PONE-D-20-20058)

Editor Comments:

We thank the Editor for the opportunity to revise our manuscript. Below, we respond in detail with bulletpoints to each of the Editor’s comments (shown in bold).

Please address reviewer concerns regarding presentation of data in the Tables, especially Table 2 which is very difficult to read/interpret. 

• Based on both Reviewers comments on the former Table 2. We have decided to split the table into two new tables, following a suggestion from Reviewer 2. Table 3 now shows the overall awareness of the programs for the sample, the percent enrolled pre-COVID, the percent enrolled after COVID, and the change. Table 4 displays the differential effect of COVID-19 on enrollment changes between those who did versus did not experience employment reduction. We believe this table is now clearer to readers.

Regarding concerns about generalizability expressed by Reviewer 2, while the authors certainly cannot control the length of time submissions are under review, perhaps the question of generalizability (or lack thereof) could be framed within a broader context. 

• We agree this is a valid concern. We have updated our paper to address changes in context that could affect generalizability:

o “Further, data from early in the pandemic provides an important baseline for evaluating the evolving changes in program participation and hardship among vulnerable individuals.” (line 95-97)

o “Since April, the pandemic spread widely across the United States, including to many southern and midwestern states with relatively weaker safety nets. However, data from April provides an important baseline to assess the evolving need of affected Americans, and to ultimately assess economic recovery.” (lines 304-307)

Finally, further discussion about challenges accessing safety net services during a pandemic could be more fully fleshed out in the revised manuscript. 

• We agree. Accordingly, we have added text as follows:

o “It is important to consider how individuals who are enrolling in safety net programs are accessing services during a pandemic. Given the closure of many places of business, individuals may be challenged to visit social services agencies for enrollment or customer service. While some customer services were expanded by phone and online, there have been widespread challenges enrolling in programs like unemployment insurance.[27] The lower than expected initial enrollment in programs like Medicaid may also reflect the challenges of navigating eligibility and enrollment processes since the pandemic.” (lines 294-300)

• We have consulted these formatting guidelines and we are now consistent with the guidelines.

2. Please include additional information regarding the survey or questionnaire used in the study and ensure that you have provided sufficient details that others could replicate the analyses. For instance, if you developed a questionnaire as part of this study and it is not under a copyright more restrictive than CC-BY, please include a copy, in both the original language and English, as Supporting Information. Moreover, please include more details on how the questionnaire was pre-tested, and whether it was validated; and clearly report the number of respondents and the response rate.

• To provide more transparency for readers, we now include the survey instrument as a supplemental.

• We have clarified that the survey instrument was not pre-tested: “The items that were used in the survey were developed by our team for the purpose of this study; items were not piloted before being used in the study.” (lines 136-137) 

• We have provide details on response rate: “The overall response rate for the panel is about 34.0% (American Association for Public Opinion Research [AAPOR] response rate three).[14]” (lines 111-112)

3. Please correct your reference to "p=0.000" to "p<0.001" or as similarly appropriate, as p values cannot equal zero.

• We have replaced these to now read “p<0.001”.

4. In your Methods section, please provide additional information about the participant recruitment method and the demographic details of your participants. Please ensure you have provided sufficient details to replicate the analyses such as: a) the recruitment date range (month and year), b) a description of any inclusion/exclusion criteria that were applied to participant recruitment, c) a table of relevant demographic details, d) a statement as to whether your sample can be considered representative of a larger population, e) a description of how participants were recruited, and f) descriptions of where participants were recruited and where the research took place.

• We have updated our methods section to provide greater detail about participant recruitment and demographics:

o a) the recruitment date range (month and year): “For this study, our team developed the State Health Access Data Assistance Center (SHADAC) COVID-19 Safety Net Survey and contracted with NORC to add the survey questions to the survey that was in the field April 23 to April 27, 2020.” (lines 120-122)

o b) a description of any inclusion/exclusion criteria that were applied to participant recruitment: “We contracted with NORC to administer the survey to a target of 1,000 respondents. The study was restricted to people over age 18. The final sample included 1,007 adults.” (lines 123-124)

o c) a table of relevant demographic details AND d) a statement as to whether your sample can be considered representative of a larger population: “Table 1 shows the demographic characteristics of the study sample. NORC develops weights to national census benchmarks and balances by gender, age, education, race/ethnicity, and region. The weighted sample is similar to a national sample of adults: 51.4% of the sample was female, 44.8% between age 18 and 44, 37.4% non-white, 46.1% with a chronic condition, 36.2% with a high school degree or less, and 83.8% residing in metropolitan areas.” (lines 124-129)

o e) a description of how participants were recruited AND f) descriptions of where participants were recruited and where the research took place: “The AmeriSpeak panel is recruited using stratified, address-based sampling methods that cover approximately 97.0% of all residential addresses. The multi-stage probability sample is created using a national frame area where blocks are sampled from within defined metropolitan or rural areas. AmeriSpeak oversamples in areas with a higher concentration of young adults and minorities and engages in additional efforts to follow up with households that initially do not respond. Individuals are recruited to the panel using a combination of US mail, telephone interviews, and in-person field interviews. Households can respond to the survey by internet (including on smartphones) or by telephone interview. About 85% of the interviews are completed online and 15% are conducted over the phone. The phone option is offered to allow “net-averse” households to participate.” (lines 108-117)

5.We note that you have stated that you will provide repository information for your data at acceptance. Should your manuscript be accepted for publication, we will hold it until you provide the relevant accession numbers or DOIs necessary to access your data. If you wish to make changes to your Data Availability statement, please describe these changes in your cover letter and we will update your Data Availability statement to reflect the information you provide.

• As earlier indicated, we will place the data in repository at acceptance and will provide the DOI to access the data.

6.PLOS requires an ORCID iD for the corresponding author in Editorial Manager on papers submitted after December 6th, 2016. Please ensure that you have an ORCID iD and that it is validated in Editorial Manager. To do this, go to ‘Update my Information’ (in the upper left-hand corner of the main menu), and click on the Fetch/Validate link next to the ORCID field. This will take you to the ORCID site and allow you to create a new iD or authenticate a pre-existing iD in Editorial Manager. Please see the following video for instructions on linking an ORCID iD to your Editorial Manager account: https://www.youtube.com/watch?v=_xcclfuvtxQ

• ORCID iD for the corresponding author, Brendan Saloner is now provided: https://orcid.org/0000-0001-9013-3023

Reviewer 1

We thank Reviewer 1 for detailed comments. Below, we respond in detail with bulletpoints to each of the Reviewer’s comments (shown in bold).

This study reports the results of a national survey conducted in April 2020 to assess the economic conditions of individuals with job loss related to the COVID-19 pandemic. This is a remarkably fast turnaround for such data and the authors are to be commended for their ability to implement these questions in the field on such a short timeline. The results are interesting, but there presentation could be clearer, and there are interesting discussion points that were not address.

• Thank you for these encouraging words and for the comments, which we believe have strengthened the clarity and depth of the paper.

Major comments:

1. Table 2 is the potentially most impactful table in the manuscript but it is difficult to interpret. I would recommend three sections in the table: 1) Enrolled before the pandemic; 2) Enrolled after the pandemic; 3) Change in enrollment. As it currently stands, it’s very difficult to understand whether the enrolled before and applied/enrolled since the pandemic are additive.

• We agree. Based on your comments and those of Reviewer 2, we have split Table 2 into 2 tables, which are now Tables 3 and 4. Table 3 shows the categories you suggested for the full sample. Table 4 shows the differential changes since the pandemic for the group that experienced employment reduction, similar to the diff-in-diff you suggested below in “minor comments.” 

2. Discussions points: Even before the pandemic, those who experienced job loss were more likely to need help from a safety net program. This speaks to the vulnerability of those impacted by COVID-19 and should be highlighted. Further, only 6.6% of individuals with job loss enrolled in Medicaid. I would expect it to be higher during a public health crisis and may speak to serious challenges in accessing public services during the pandemic.

• In the Discussion, we now highlight the point that those who experienced job loss were more likely to need help from a safety net program prior to the pandemic: “Overall, our study findings highlight the precarity of individuals who were most affected early in the COVID-19 pandemic” (lines 213-214)

• We also more explicitly address the issue of low initial uptake of Medicaid early in the pandemic and how this might reflect challenges accessing public services: “Lower than expected initial enrollment in programs like Medicaid may also reflect the challenges of navigating eligibility and enrollment processes since the pandemic.” (lines 296-298)

3. The last sentence of abstract could be interpreted as individuals are “stretching the safety net programs” but this is what the programs were designed to do. Rephrase to indicate that safety net systems will remain critical until the pandemic is over and should be protected from funding cuts. Please correct similar language in the discussion. For example: “Food assistance programs are of particular concern” - these programs not “of concern” but I do agree that their stability is imperative. The problem is not that they are being used - it’s that the US didn’t develop the programs in a way that could respond to large recessions/depressions. A well run program would be able to support individuals during downturns in the economic cycle.

• We strongly agree and regret that our prior phrasing did not emphasize the importance of stabilizing safety net programs to meet the needs of vulnerable people. We have rephrased in several places. 

o For example, last line of abstract: “The economic devastation from COVID-19 increases the importance of a robust safety net.” (lines 50-52)

o In the Discussion: “The stability of food assistance programs is of particular concern.” (line 237)

o “The affordability of housing is another concern.” (line 250)

Minor comments:

1. Rather than comparing those with job loss to those without job loss who enrolled, would be more interesting to highlight how much those with job loss contributed to increases in use of safety net services. This could be done with a simple diff-in-diff model to determine how much of the increase in safety services pre and during COVID was associated with job loss.

• Consistent with your proposed modification to Table 2, we have created Table 4, which has the layout that captures this “difference-in-differences” style effect.

2. To build on this comment, comparisons between those with and without covid job loss are not wrong but are somewhat odd. Of course those who lost their job would rely on safety net health systems more than individuals who did not lose their jobs. If the authors believe this is the most appropriate comparison in the data, then a clear justification for why such a comparison is important is warranted.

• Thanks for pushing us to clarify this. Our point is consistent with your suggestion #1: to demonstrate the disproportionate need among this group. We state this more clearly in the manuscript:

o “Because we were interested in identifying the disproportionate changes in program participation among those experiencing employment reduction, we fit a regression model for program participation that estimates the average change since the pandemic, the baseline rate for those without employment reduction, and an interaction term. This interaction term is analogous to a difference-in-differences coefficient, representing the change in program participation since the pandemic for those with employment reduction versus those who without employment reduction.” (lines 150-156)

3. Lines 147-152 - The first part of the sentence refers to those with and without job loss. Unclear whether the later part of the sentence related to general population or only those with job loss. Please clarify.

• We have clarified this section of the paper.

4. There is substantial commentary on Medicaid in the discussion, but would shorten and refocus on what can be gleaned from the study’s findings. For example, what does it mean for budgets, health care delivery, health system budgets, etc that millions more individuals will be relying on Medicaid?

• We have reduced the discussion related to the specific provisions of the Medicaid program and have refocused on the study findings as follows: 

o “The increased enrollment in public programs could have important implications for state budgets and delivery system. Early in the pandemic, state Medicaid programs began projecting that the weakened economy would increase program enrollment and total spending.[23] Testing and treatment related to COVID-19 could also contribute to rising Medicaid spending, though this spending could be offset by other forms of medical care that decreased after the pandemic, such as preventive office visits. State programs have a variety of policy options for easing transitions of new members into Medicaid, reducing churn, and simplifying enrollment. A recent analysis suggests that Medicaid enrollment increased early in the pandemic, although these increases were not correlated with enrollment changes in those states [24]” (lines 275-283)

 

Reviewer 2

We thank Reviewer 2 for detailed comments. Below, we respond in detail with bulletpoints to each of the Reviewer’s comments (shown in bold).

This paper is a straightforward comparison of economic hardship and enrollment in safety net programs between those who experienced COVID-19-related job loss (defined as either complete job loss or a reduction in hours) versus those who didn’t. Data come from a special COVID-19 economic impact questionnaire that the authors had appended to the April 24-26 survey of the AmeriSpeak panel. The key results are: 1) 28% of the sample experienced job loss, 2) there were sizeable differences in program participation even before the pandemic between those who would eventually experience job loss and those who would not (i.e. job loss hit the lower end of the labor market harder), and 3) those differences became even larger after the job loss happened, 4) there were huge differences in subsequent priorities for stimulus check spending and ability to pay for basic needs. There is also an attempt to examine the influence of demographic characteristics on these results, though the estimates are generally too imprecise to be useful. 

• Thank you for this helpful synopsis of our paper which we agree touches on key points of interest.

These results are unsurprising and the analyses are fairly simplistic (e.g. no attempt to leverage exogenous variation to identify the causal effects of job loss on the various outcomes, rather than just its associations). That said, the data source is novel and there is certainly value to quantifying the magnitudes of the associations. The more attention that can be brought to the need for a robust social safety net and continued stimulus during this time of crisis, the better.

• Thank you, we agree that the paper is entirely descriptive and that we do not exploit exogenous variation. Per your comment, our main intention in this paper is to demonstrate the magnitudes of need for services early in the pandemic. We have added a suggestion in the paper for further work to examine exposure to COVID-19 cases:

o “Future research, with larger sample sizes, could beneficially compare differences in program participation based on residence in areas experiencing higher COVID-19 case rates.[28]” (lines 308-310)

Comments/suggestions:

1) I don’t see the value of the regressions in Table 5. As the authors themselves concede, the confidence intervals tend to be too wide for the results to be terribly informative. I would just drop this table, freeing up space for better exposition in the text and potentially also allowing the clunky Table 2 to be split into multiple exhibits (more on that next).

• We agree that these results are not that informative because of the imprecision and have removed the regression table.

2) It took me a while to figure out exactly what’s going on in Table 2. The confusing part is that there’s really two very distinct questions being asked: 

a) How different was baseline enrollment (i.e. pre-job-loss) between those who would eventually lose their jobs/have hours cut versus those who would not? If I understand correctly, the value of those results is in showing that the people who got laid off/had their hours cut were disproportionately “vulnerable” even prior to the pandemic. 

b) How different was new enrollment during the pandemic between those who experienced job loss/reduction versus those who did not? This is more like the usual difference-in-difference-style question.

Since these are such distinct questions, I think it would be less confusing if they were treated more distinctly throughout the paper. For instance, split Table 2 into two separate tables, and better explain their distinctiveness in the abstract, intro, and body of the paper. To provide one example of how the distinction is currently murky, the abstract says, “Those who experienced COVID-19 job loss versus those who did not were significantly more likely to have applied or enrolled in >1 program … and also significantly more likely to specifically have enrolled in unemployment insurance … and SNAP.” More likely to have enrolled when? Nothing in that sentence specifies new enrollment during the pandemic. So it ends up reading as a continuation of the previous sentence about baseline differences. 

• Thank you for these helpful interpretative points, which dovetail with concerns expressed by Reviewer 1. We regret that the table was confusing in the prior draft. Based on your suggestions, we have split the former Table 2 into two pieces: Table 3 that shows the overall rates of program participation before the pandemic, since the pandemic, and the change and Table 4, which represents the difference-in-differences, and is more about the contrast between those who experienced and did not experience employment reduction.

3) One substantive critique is that the authors ignore the well-known issue of misreporting in survey-based program participation measures. While the critique applies to all programs, the literature on misreporting in SNAP is especially well-developed. A recent paper that provides a detailed discussion of that literature is https://onlinelibrary.wiley.com/doi/abs/10.1002/soej.12364

• Thank you for raising this very reasonable issue. This reporting bias is likely to lead us to understate program participation overall, but we have no a priori reason to believe that such bias is likely to be greater among people with versus without COVID-19 related job loss. We added further context on this and included the helpful suggested citation: 

o “Self-reported program participation may be particularly problematic for some programs such as SNAP,[30] although we have no reason to believe that reporting bias will be differential between people who experienced COVID-19 employment loss versus those that did not.” (lines 311-314)

4) Given the rapidly changing nature of the pandemic, a survey from late April is already a bit dated. The authors acknowledge that “data collected in late April 2020 may not generalize to issues and concerns arising in more recent periods.” I appreciate the caveat but they should push harder here. If the results don’t generalize, then why do we care? At lease some argument needs to be made for generalizability.

• Very fair point. We expand on this issue in the paper, arguing that the early experience in the pandemic can offer an important baseline for further study. The areas that were hardest hit during this time were northeast states with stronger safety net programs, whereas the pandemic in summer 2020 evolved to more directly affect areas with weaker safety net programs. We have added additional text on this issue:

o “Further, data from early in the pandemic provides an important baseline for evaluating the evolving changes in program participation and hardship among vulnerable individuals.” (line 95-97)

o “Since April, the pandemic spread widely across the United States, including to many southern and midwestern states with relatively weaker safety nets. However, data from April provides an important baseline to assess the evolving need of affected Americans, and to ultimately assess economic recovery.” (lines 304-307)

5) p. 13: “There was no statistically significant difference in participation in at least one safety net program between people who subsequently experienced COVID-19 job loss than those not experiencing job loss (50.0% versus 37.7%, p=0.120).” I don’t like this sentence. The magnitude of that difference is large, and the p-value is close to the threshold … more nuance is needed. 

• Yes, we agree that the prior statement was overly focused on the p-value. However, we are no longer making this comparison in the paper, so this has been removed.

6) I’d give some thought to whether “job loss” is the best term to describe the “treatment” group. It’s actually the combination of those who lost jobs completely and those who had their hours reduced – and the majority is the latter. It’s really more of an “adverse employment shock” group. Calling it “job loss” may be catchier but it is a recipe for being misquoted in the media. 

• This is an astute observation. We have renamed this group “employment reduction” and more clearly indicated what we mean by this term.

---

## [Decision Letter · Decision Letter 1]

21 Sep 2020

Access and enrollment in safety net programs in the wake of COVID-19: A national cross-sectional survey

PONE-D-20-20058R1

Dear Dr. Saloner,

We’re pleased to inform you that your manuscript has been judged scientifically suitable for publication and will be formally accepted for publication once it meets all outstanding technical requirements.

Kind regards,

Nickolas D. Zaller

Academic Editor

PLOS ONE

Additional Editor Comments (optional):

Reviewers' comments:

Reviewer's Responses to Questions

**Comments to the Author**

1. If the authors have adequately addressed your comments raised in a previous round of review and you feel that this manuscript is now acceptable for publication, you may indicate that here to bypass the “Comments to the Author” section, enter your conflict of interest statement in the “Confidential to Editor” section, and submit your "Accept" recommendation.

Reviewer #1: All comments have been addressed

Reviewer #2: All comments have been addressed

2. Is the manuscript technically sound, and do the data support the conclusions?

Reviewer #1: Yes

Reviewer #2: Yes

3. Has the statistical analysis been performed appropriately and rigorously? 

Reviewer #1: Yes

Reviewer #2: Yes

4. Have the authors made all data underlying the findings in their manuscript fully available?

Reviewer #1: Yes

Reviewer #2: Yes

5. Is the manuscript presented in an intelligible fashion and written in standard English?

Reviewer #1: Yes

Reviewer #2: Yes

6. Review Comments to the Author

Reviewer #1: The authors have satisfactorily addressed my concerns. These data will be a nice contribution to our understanding of the economic and social impact of COVID-19.

One very minor comment - Job loss should be changed to employment reduction in Tables 5 and 6.

Reviewer #2: (No Response)

7. PLOS authors have the option to publish the peer review history of their article (what does this mean?). If published, this will include your full peer review and any attached files.

Reviewer #1: No

Reviewer #2: **Yes: **Charles J. Courtemanche

---

## [Editor Report · Acceptance letter]

28 Sep 2020

PONE-D-20-20058R1 

Access and enrollment in safety net programs in the wake of COVID-19: A national cross-sectional survey 

Dear Dr. Saloner:

I'm pleased to inform you that your manuscript has been deemed suitable for publication in PLOS ONE. Congratulations! Your manuscript is now with our production department. 

Kind regards, 

on behalf of

Dr. Nickolas D. Zaller 

Academic Editor

PLOS ONE